The complete chloroplast genomes of seventeen Aegilops tauschii: genome comparative analysis and phylogenetic inference

Su Qing 1
Liu Luxian 1
Zhao Mengyu 1
Zhang Cancan 1
Zhang Dale 1
Li Youyong liyouyong@163.com 2
Li Suoping lisuoping@henu.edu.cn 1
1 Key Laboratory of Plant Stress Biology, School of Life Science, Henan University , Kaifeng , China
2 School of Life Science and Technology, Henan Institute of Science and Technology , Xinxiang , China
Lazo Gerard
Electronic publication date: 2020 Mar 4
Publication date: 2020
Volume: 8
Electronic Location ID: e8678
Received 2019 Aug 21; Accepted 2020 Feb 3
Copyright: ©2020 Su et al.
Copyright year: 2020
Copyright holder: Su et al.
License: This is an open access article distributed under the terms of the Creative Commons Attribution License, which permits unrestricted use, distribution, reproduction and adaptation in any medium and for any purpose provided that it is properly attributed. For attribution, the original author(s), title, publication source (PeerJ) and either DOI or URL of the article must be cited.
License URL: https://creativecommons.org/licenses/by/4.0/

Keywords: Chloroplast genome, Aegilops tauschii, Common wheat D progenitor, Genome characteristic, Genome comparative, Phylogenetic analysis, cpDNA markers, Genetic differentiation, Next-generation sequencing, The spreading route

Funding: National Natural Science Foundation of China 31601297 31871615 Key Scientific Research Projects of Higher Education Institutions in Henan Province 20A210007 This work was supported by the National Natural Science Foundation of China (Grant Nos. 31601297 and 31871615) and the Key Scientific Research Projects of Higher Education Institutions in Henan Province (Grant No. 20A210007). The funders had no role in study design, data collection and analysis, decision to publish, or preparation of the manuscript.

==============================
The D genome progenitor of bread wheat, Aegilops tauschii Cosson (DD, 2n = 2x = 14), which is naturally distributed in Central Eurasia, ranging from northern Syria and Turkey to western China, is considered a potential genetic resource for improving bread wheat. In this study, the chloroplast (cp) genomes of 17 Ae. tauschii accessions were reconstructed. The cp genome sizes ranged from 135,551 bp to 136,009 bp and contained a typical quadripartite structure of angiosperms. Within these genomes, we identified a total of 124 functional genes, including 82 protein-coding genes, 34 transfer RNA genes and eight ribosomal RNA genes, with 17 duplicated genes in the IRs. Although the comparative analysis revealed that the genomic structure (gene order, gene number and IR/SC boundary regions) is conserved, a few variant loci were detected, predominantly in the non-coding regions (intergenic spacer regions). The phylogenetic relationships determined based on the complete genome sequences were consistent with the hypothesis that Ae. tauschii populations in the Yellow River region of China originated in South Asia not Xinjiang province or Iran, which could contribute to more effective utilization of wild germplasm resources. Furthermore, we confirmed that Ae. tauschii was derived from monophyletic speciation rather than hybrid speciation at the cp genome level. We also identified four variable genomic regions, rpl32-trnL-UAG, ccsA-ndhD, rbcL-psaI and rps18-rpl20, showing high levels of nucleotide polymorphisms, which may accordingly prove useful as cpDNA markers in studying the intraspecific genetic structure and diversity of Ae. tauschii.

Introduction

Aegilops tauschii Cosson (DD, 2n = 2x = 14), which is the D genome progenitor of common bread wheat, is characterized by abundant genetic variation and is noted for its strong tillering ability and high plant tolerance (resistance to disease, drought and abiotic stresses) (Singh et al., 2012). Ae. tauschii is widely distributed in Central Eurasia, ranging from northern Syria and Turkey to western China (the Yili Area of Xinjiang). In addition, as a farmland weed that co-occurs with common wheat, Ae. tauschii is also found in the middle reaches of the Yellow River (including Henan and Shannxi provinces, China) (Wei et al., 2008). On the basis of the findings of genetic studies, Ae. tauschii has been divided into two sublineages based on nuclear genome sequences, namely L1 and L2, which are broadly affiliated with Ae. tauschii ssp. strangulata and Ae. tauschii ssp. tauschii, respectively (Mizuno et al., 2010; Dvorak et al., 1998). Previous studies have proved that L2 lineage is involved in the origin of common wheat and is limited to a narrow area within the whole species’ overall distribution range (Wang et al., 2013; Dvorak et al., 2012), as well as L1 lineage is adapted to more diverse environments (Dudnikov, 2012). Given the large genetic distance between L1 and L2, it has been proposed that Ae. tauschii (especially L1 lineage) showes more genetic differentiation than does the D genome of the common wheat (Lubbers et al., 1991; Wang et al., 2013; Dvorak et al., 2012). Thus, in common with wild crop progenitors, Ae. tauschii, especially L1 lineage, is considered to represent a potentially valuable germplasm resource that could be exploited for genetic improvement of common wheat (Kilian et al., 2011; Zhang et al., 2017; Zhang et al., 2018). Moreover, owing to widely geographical distribution of the L1 lineage, studying of its genetic and evolutionary relationships contribute to more effective utilization of wild germplasm resources.

Iran is widely regarded as the center of the origin and genetic diversity of Ae. tauschii (Dvorak et al., 1998). However, as a consequence long periods of dispersal and adaptation, this species now shows distinct difference within its distribution range. At present, L2 lineage is mainly restricted to an area extending from Transcaucasia (Armenia and Azerbaijan) to eastern Caspian Iran, whereas L1 lineage is more widely distributed across the entire species range, which includes the middle reaches of the Yellow River and the Yili area of Xinjiang Province in China (Kihara, Yamashita & Tanaka, 1965; Jaaska, 1980; Matsuoka, Takumi & Kawahara, 2015; Wei et al., 2008). As a clade of L1 lineage, there is still some debate regarding the specific spreading route from Xinjiang to the middle reaches of the Yellow River in China, given the large geographical isolation of these two areas. Some researchers believe that the long-distance spread occurred concomitantly with the expansion of original wheat varieties and Ae. tauschii accessions collected from the Yellow River are introduced from Xinjiang Province (Wang et al., 2010; Yen, Cui & Dong, 1984). However, on the basis of the established genetic similarities among 31 Ae. tauschii accessions from China and Iran determined using SSR markers, Wei et al. (2008) proposed that Ae. tauschii populations in the Yellow River region are likely to have been directly introduced from Iran along the silk road, which indicates a longer genetic relationship with Xinjiang landraces. In addition, Mizuno et al. (2010) used AFLP molecular markers to classify genetic types among Ae. tauschii accessions and found that the accessions collected in the Yellow River basin, which were significantly different from the L1E accessions in Afghanistan and Xinjiang, closely clustered to the L1W accessions in the Middle East.

With regard to the introduction of Ae. tauschii in China, it is believed that the routes by which barley and wheat spread from western Eurasian to East Asia may offer some relevant clues. One hypothesis postulated that common wheat arrived from the west (probably from Afghanistan or the Central Asia oases), moving eastwards into northern Xinjiang (Betts, Jia & Dodson, 2014; Wu et al., 2019), whereas an alternative opinion envisaged that common wheat may have reached East Asia along a north-west passage from Eurasia, via southern Siberia and Mongolia. In both scenarios, common wheat was assumed to have initially spread into Central Gansu and thereafter migrated eastwards into the Yellow River region or was later introduced to the margins of the Qinghai–Tibetan Plateau in highland China (Dodson et al., 2013; Betts, Jia & Dodson, 2014; Long et al., 2018; Wu et al., 2019). A subsequent dispersal route from the margins of the Qinghai–Tibetan Plateau to the Yangtze Valley has also been proposed (Wu et al., 2019). It is generally considered that barley had arrived to the northeastern and southeastern Tibetan Plateau at a date prior to 4,000 calendar years ago (Wu et al., 2019). However, in contrast to the aforementioned traditional views regarding the dispersal routes of common wheat and barley (Dodson et al., 2013; Betts, Jia & Dodson, 2014; Long et al., 2018), Zeng et al. (2018) have suggested that qingke barley is derived from eastern domesticated barley and was introduced from South Tibet, most likely from northern Pakistan, India, and Nepal eastwards into the Tibetan Plateau, which is supported by recent archaeological evidence of the occurrence of barley in north-east India. Thus, the specific route whereby Ae. tauschii extended its range is still ambiguous, and remains considerable interest for intensive exploration.

Owing to the notable prominent advantages of the cp genome, including a moderate rate of nucleotide replacement, significant variations in the rate of molecular evolution between non-coding and coding regions, moderate genome size, and desirable collinear properties among different species (Liu et al., 2018), an analysis of cp genome sequences is considered an effective strategy for investigating intra- and interspecific evolutionary relationships, as well as being of considerable utility in comparative genomic studies (Matsuoka, Mori & Kawahara, 2005; Yamane & Kawahara, 2005; Tabidze et al., 2014; George et al., 2015; Liu et al., 2017; Shang et al., 2019). In angiosperms, the size of cp genomes and their gene arrangements are generally highly conserved with a circular chromosome ranging in size from 120 to 160 kb, and comprising a small single-copy (SSC) region, a large single-copy (LSC) region and a pair of inverted repeats (IRs) region (Palmer, 1991; Yang et al., 2010). Given that cp are primarily non-recombining and uniparentally inherited, phylogenetic analysis based on cp sequences can also facilitate specific identification of maternal lineages (Sang, 2002).

Using complete cp genomes, a number of previous studies have examined genetic variations in common wheat and its relatives and performed related phylogenetic analyses (Middleton et al., 2014; Gornicki et al., 2014; Gogniashvili et al., 2016). In the present study, with a view toward sheding light on the genetic variation of Ae. tauschii and the source of Chinese landraces, we performed sequence analysis of the complete cp genomes of 17 Ae. tauschii accessions derived from regions spanning the known distribution range from western Turkey to eastern China. The results not only provided a series of new insights to clarify the spreading route of Ae. tauschii, but also enabled us to identify promising germplasm resources for the genetic improvement of bread wheat.

Material and Methods

Plant materials

Information related to the 17 Ae. tauschii accessions examined in this study, including origin and collection region, are listed in Table 1. The accessions marked as ‘XJ’, ‘T’, and ‘S’ are those from Xinjiang, Henan, and Shannxi Provinces, respectively. These materials were collected in the field by Key Laboratory of Plant Stress Biology of Henan University. The seeds with the accessions designated as ‘AY’ and ‘AS’ were provided by the US National Plant Germplasm Center and Institute of Genetics and Developmental Biology. The sources of these accessions are distributed across a wide geographical range that extends from Turkey, Georgia, Iran, Turkmenistan, Kazakhstan, Tajikistan, Afghanistan, Pakistan, and India to Xinjiang, Shannxi and Henan Provinces in China (Fig. 1), and comprises 15 lineage L1 accessions and two lineage L2 accessions, as determined based on single-nucleotide polymorphisms (SNPs) (Wang et al., 2013).

Table 1 Origin and collection regions of 17 Aegilops tauschii accessions.

Accessionsa	Inventoryb	Source	Regions	
SC1		Shannxi, China	N (34.158997), E (108.90699), Elevation: 428 m	
AY81	PI 542277	Izmir, Turkey	Elevation: 30 m	
AY34	PI 662116	Khujand, Tajikistan	N (39.771944), E (68.809444), Elevation: 433 m	
AY22	PI 511365	Baluchistan, Pakistan	N (30.925), E (66.44638889), Elevation: 675 m	
AY320	PI 554324	Kars, Turkey	N (40.13333333), E (43.06666667), Elevation: 1,275 m	
AY21	PI 662091	Khujand, Tajikistan	N (40.67388889), E (70.54694444), Elevation: 462 m	
XJ04		Xinjiang, China	N (44.321239), E (80.77766), Elevation: 892 m	
XJ0109		Xinjiang, China	N (43.386026), E (83.5977), Elevation: 1,269 m	
T093		Henan, China	N (35.728123), E (115.242698), Elevation: 52 m	
XJ098		Xinjiang, China	N (43.386026), E (83.5977), Elevation: 1,269 m	
AY78	PI 210987	Kondoz, Afghanistan	N (36.68333333), E (68.05), Elevation: 362 m	
AS060	PI 511369	Iran	N(36.85), E(55.170593), Elevation: 527 m	
AY48	PI 603225	Turkmenistan, Balkan	N(38.48333333), E(56.3), Elevation: 730 m	
AY076	PI 276975	Turkistan	N (45), E (70), Elevation: 210 m	
AY20	PI 574469	India	N(20), E(77), Elevation: 509 m	
AY72	PI 428563	Georgia	N (43),E (47) , Elevation: 90 m	
AY46	PI 511368	Tehran, Iran	N(35.8), E(50.96666667), Elevation: 1,296 m	
Notes.

a Accession numbers from Key Laboratory of Plant Stress Biology of Henan University.

b Inventory provided by the US National Plant Germplasm Center and Institute of Genetics and Developmental Biology.

Figure 1 Geographical distribution of 17 Aegilops tauschii accessions from western Turkey to eastern China.

Next-generation sequencing, annotation and comparsion of chloroplast genomes

For each of the 17 Ae. tauschii accessions, total genomic DNA was extracted from fresh leaves of 1-week-old seedlings germinated by the aboved seeds of Ae. tauschii by the method of Han et al. (2015). Approximately 5 to 10 µg of extracted DNA was sheared to generate fragments, and the quality of DNA sequences was determined using an Agilent Bioanalyzer 2100 (Agilent Technologies). Thereafter, we generated a paired-end sequencing library, which was constructed from ∼400 bp fragments obtained using a Genomic DNA Sample Prep Kit (Illumina) in accordance with the manufacturer’s protocol. Subsequent genome sequencing was carried out by Majorbio Bio-Pharm Technology Co., Ltd. (Shanghai, China) using the HiSeq X ten sequencing platform (Illumina Inc., San Diego, CA) with 150 bp read length. Low quality reads with a phred score <30 and 0.001 probability error were removed using Trimmomatic v0.36 (http://www.usadellab.org/cms/index.php?page=trimmomatic), and the remaining high quality fragment s were assembled into contigs using the SOAPdenovo v2.21 (http://soap.genomics.org.cn/soapdenovo.html). The assembled contigs were further aligned to the AL8/78 reference genome (GenBank No. KJ 614412) using BLASTN (http://www.ncbi.nlm.nih.gov). Finally, gaps in the genome sequence were filled using GapCloser v1.12 (http://soap.genomics.org.cn/soapdenovo.html).

The complete cp genome of Ae. tauschii was annotated using the Dual Organellar GenoMe Annotator program (DOGMA, Wyman, Jansen & Boore, 2004), and then verified artificially. The annotation of the tRNA genes was verifed using tRNAscan-SE. The circular cp genome of Ae. tauschii was constructed using OGDRAW online software (Lohse et al., 2013). The Gene Ontology (GO) functional categories were execute d using Blast2GO V3.2 software.

In order to determine intraspecific variations, variations in the LSC, SSC region, and IR regions, we used Geneious 9.0.5 software (Biomatters, Auckland, New Zealand). Using AL8/78 as the reference genome, the cp genomes of T093 and AY81 were performed to visualize the sequence variations by mVISTA.

Phylogenetic analysis

The phylogenetic relationships among Ae. tauschii accessions were examined using the complete cp genome sequences of the 17 accessions. In addition, in order to establish the origin of Ae. tauschii within the family Triticeae, we performed phylogenetic analysis using the genome sequences of the following 99 accessions: the 17 accessions newly sequenced in the present study, 56 accessions in the genus Aegilops, 24 accessions in the genus Triticum, and two accessions of Hordeum vulgare as the outgroups.

The sequences of these cp genomes were aligned using Geneious 9.0.5 software (Biomatters, Auckland, New Zealand). Gaps were adjusted manually or removed. The alignment lengths with all gap positions removed were determined to be 135,984 bp (17 accessions) and 131,116 bp (99 accessions). We constructed two corresponding phylogenetic trees based on the maximum-likelihood (ML) method utilizing MEGA7.0 software (Kumar, Stecher & Tamura, 2016). The bootstrap values were evaluated with 1000 replications. The best-fit nucleotide substitution model (GTR+I+G) was selected using jModelTest 2.1.4 software (Posada, 2008) operating with default parameters.

Analysis of sequences divergence and molecular markers

An analysis of cp genome divergence among the 17 Ae. tauschii accessions was performed by employing MAFFT alignment based on homologous genomic regions (Katoh et al., 2002). Nucleotide diversity (Pi) of polymorphic sequences among the 17 Ae. tauschii individuals were calculated utilizing DnaSP v5.0 software (Librado & Rozas, 2009).

Results

Characteristics and comparison of Ae. tauschii cp genomes

The cp genomes of the 17 Ae. tauschii accessions sequences in the present study were found to be similar with respect to size and gene content. They are generally 135,551∼136,009 bp in length and contained four quadripartite structure: two IR regions separated by a SSC region (12,773∼12,826 bp) and a LSC region (79,723∼80,140 bp) (Fig. 2 and Table 2). The intergenic regions of these sequences ranged within 75,635∼77,699 bp in length, accounting for 53.2% to 57.2% of the total genomes. The remaining sequence gene regions comprise coding regions, intron regions, tRNA genes, and rRNA genes. Notably the GC contents of the 17 cp genomes are relatively stable ranging from 38.31% to 38.35% (Table 2).

Figure 2 Chloroplast genome map of 17 Aegililops tauschii accessions.

The genes lying outside and inside of circle are transcribed in the counterclockwise and clockwise directions, respectively. Different colored bars are used to represent different functional gene groups. The darker gray area in the inner circle displays GC content. The fine lines indicate the boundary of the inverted repeats (IRA and IRB) that split the genomes into large single copy (LSC) and small single copy (SSC) regions.

Table 2 Comparative analysis of the chloroplast genomes among 17 Aegilops tauschii accessions.

Accessions	AY21	AY22	AY320	AY34	AY81	SC1	T093	XJ04	XJ098	XJ0109	AY78	AS060	AY20	AY72	AY48	AY076	AY46	
Total size (bp)	135,858	136,009	135,978	135,777	135,890	135,608	135,850	135,610	135,611	135,613	135,656	135,634	135,617	135,813	135,836	135,854	135,551	
GC%	38.32	38.31	38.33	38.32	38.31	38.33	38.32	38.33	38.33	38.33	38.32	38.32	38.33	38.33	38.31	38.33	38.35	
Gene total length (bp)	59,706	59,703	59,559	59,703	59,628	59,703	63,575	58,583	59,703	57,914	59,703	59,703	59,703	59,934	59,916	59,916	59,916	
Gene average length (bp)	719	719	717	719	718	719	722	709	719	699	719	719	719	722	721	721	721	
Gene density (genes per kb)	0.61	0.61	0.61	0.611	0.61	0.612	0.647	0.61	0.612	0.611	0.611	0.611	0.612	0.611	0.611	0.61	0.612	
GC content in gene region (%)	38.9	38.9	38.9	38.9	38.9	38.9	38.8	38.8	38.9	39.1	38.9	38.9	38.9	38.9	38.9	38.9	38.9	
Gene/Geonme (%)	43.9	43.9	43.8	44	43.9	44	46.8	43.3	44	42.8	44	44	44	44.1	44.1	44.1	44.2	
Intergenic region length (bp)	76,152	76,306	76,419	76,074	76,262	75,905	72,275	77,027	75,908	77,699	75,953	75,931	75,914	758,79	75,920	75,938	75,635	
GC content in intergenic region (%)	37.8	37.8	37.8	37.8	37.8	37.8	37.8	37.9	37.8	37.7	37.8	37.8	37.8	37.8	37.8	37.8	37.8	
Intergenic length/Genome (%)	56.1	56.1	56.2	56	56.1	56	53.2	56.7	56	57.2	56	56	56	55.9	55.9	55.9	55.8	
LSC (bp)	79,991	80,142	80,111	79,910	80,020	79,741	79,983	79,744	79,743	79,746	79,789	79,766	79,749	79,946	79,969	79,987	79,723	
SSC (bp)	12,771	12,771	12,771	12,771	12,774	12,771	12,771	12,771	12,772	12,771	12,771	12,772	12,772	12,771	12,771	12,771	12,732	
IR (bp)	21,548	21,548	21,548	21,548	21,548	21,548	21,548	21,548	21,548	21,548	21,548	21,548	21,548	21,548	21,548	21,548	21,548	
Total number of genes	124	124	124	124	124	124	124	124	124	124	124	124	124	124	124	124	124	
Number of protein-coding genes	82(6)	82(6)	82(6)	82(6)	82(6)	82(6)	82(6)	82(6)	82(6)	82(6)	82(6)	82(6)	82(6)	82(6)	82(6)	82(6)	82(6)	
Number of rRNA genes	8(4)	8(4)	8(4)	8(4)	8(4)	8(4)	8(4)	8(4)	8(4)	8(4)	8(4)	8(4)	8(4)	8(4)	8(4)	8(4)	8(4)	
Number of tRNA genes	34(7)	34(7)	34(7)	34(7)	34(7)	34(7)	34(7)	34(7)	34(7)	34(7)	34(7)	34(7)	34(7)	34(7)	34(7)	34(7)	34(7)	
Duplicated genes in IR	17	17	17	17	17	17	17	17	17	17	17	17	17	17	17	17	17	

We also found that the compositions of 17 genomes were highly conserved, with each of the accessions containing a total of 124 functional genes (82 protein-coding genes, 8 rRNAs, and 34 tRNAs), of which 17 were duplicated genes (Table 2). Among the 107 unique genes, 57 are related to self-replication and 46 are associated with photosynthesis. The functions of the remaining four annotated genes are as follows: a maturase (matK), an envelope membrane protein (cemA), a C-type cytochrome synthase (ccsA) as well as a protease (clpP) (Table S1). As shown in Table S1, the 124 functional genes were were classified into three groups and assigned according annotations clearly. Then, the Gene Ontology (GO) functional categories among these genes were investigated using Blast2go (Conesa et al., 2005) which were mainly enriched for cellular process, metabolic process, cell, cell part, and organelle (Fig. S1 and Table S2).

The sequences of representative genomes of three Ae. tauschii accessions were relatively conserved, and any translocations or inversions among these genomes were not identified. The IR regions showed lower sequence variation than either LSC or SSC regions (Fig. S2). Furthermore, the precise IR/SSC and IR/LSC boundary locations and the corresponding neighboring genes were found to be identical in length (21,548 bp) and border structure (Fig. S3).

Phylogenetic tree

The two cp genome datasets (the 17 complete cp genome sequences assembled in this study and the 99 selected cp genome sequences) were used to reconstruct the corresponding phylogenetic trees. As shown in Fig. 3, the 17 newly sequenced Ae. tauschii accessions were clustered into three groups, with group I being a sister to group II and group III, which formed a single clade. Group I comprised two accessions from Iran and Turkey, which derived from the L2 lineage, whereas the accessions in groups II and III all originated from the L1 lineage. Further, group II included two accessions from Tajikistan, three accessions from Afghanistan, Turkey and Georgia. Group III encompassed two accessions from the Yellow River region (SC1 and T093), three accessions from Xinxiang (XJ0109, XJ04 and XJ098), and a further five accessions from India, Iran, Turkmenistan, Turkistan and Pakistan, respectively.

Figure 3 Phylogenetic tree constructed based on the complete chloroplast genomes of 17 Aegililops tauschii accessions by maximum likehood (ML) method.

Bootstrap support values (<50%) were hided. The phylogenetic tree resulting from analysis of 135,984 bp in the alignment length of chloroplast genomes with all gap positions removed, including long stretches of the same nucleotide, short sequences appearing in opposite orientation and some sequences consisting short repeats.

The phylogenetic tree of Poaceae faminly, inferred based on the cp genome sequences of 99 selected accessions, was characterized by the two genetic clusters, namely Triticum and Aegilops clusters (Fig. 4). However, we found that Ae. speltoides was clustered in a polytomy together with almost all the Triticum species, whereas Triticum urartu and Triticum monococcum were grouped with the remaining Aegilops species in the Aegilops cluster. Furthermore, all D-genome species, including Ae. tauschii, Ae. ventricosa and Ae. cylindrica, were found to cluster in a single clade.

Figure 4 Phylogenetic tree constructed based on the complete chloroplast genomes of 99 Triticeae accessions by maximum likehood (ML) method.

The phylogenetic tree resulting from analysis of 131,116 bp in the alignment length of chloroplast genomes with all gap positions removed, including long stretches of the same nucleotide, short sequences appearing in opposite orientation and some sequences consisting short repeats.

Sequence divergences of Ae. tauschii cp genomes and molecular marker development

In order to elucidate variant characteristics among the 17 Ae. tauschii cp genomes, we further analyzed divergences in the sequences of coding genes, intron regions and intergenic regions. We accordingly detected 56 variation loci among which 38 loci were located in non-coding regions (34 intergenic regions and four intron regions) and the remaining 18 loci were found in coding regions. A relatively high value of nucleotide variability (Pi) was thus determined for non-coding regions, ranging from 0.00008 to 0.00635 with an average of 0.00133, which was approximately three times greater than that in the coding regions (average: 0.000432) (Table S3). Among these variable loci, rpl32-trnL-UAG (0.00478), ccsA-ndhD (0.00483), rbcL-psaI (0.00492), and rps18-rpl20 (0.00635), which are located in intergenic regions (the former two in the SSC region and the latter two in the LSC region), displayed the highest nucleotide polymorphisms (Fig. 5). Moreover, the primers sequences of four regions can be effectively amplified after verification (Table S4), which will help for studying the intraspecific genetic structure and diversity of Ae. tauschii.

Figure 5 Comparative analysis of the nucleotide variability (Pi) values among 17 Aegililops tauschii accessions.

The homologous regions are oriented according to their locations in the chloroplast genomes.

Discussion

In this study, we sequenced the cp genomes of 17 geographically dispersed Ae. tauschii accessions, all of which display the typical angiosperms structure and harbor an identical component of 107 unigenes arrayed in the same order (Fig. 2). These genomes were found to be relatively conserved, with the IR regions showing greater conservation than either the LSC or SSC region (Fig. S1). Differences in genome size could mainly be attributed to intraspecific differences among the cp genomes rather than expansion and contraction of IR regions. We found that the size of the 17 assembled genomes tended to be similar, ranging from 135,551 bp to 136,009 bp (Table 2), with accessions AY22 and AY46 having the largest and the smallest genomes, respectively. The IRs regions of all 17 genomes were identical in length (21,548 bp) and differences in genome sequence length can largely be attributed to the variation in the non-coding regions, particularly with respect to the size of the intergenic regions size (Table 2). Comparative analysis with the reference genome indicated that there has been no loss of genes in any of the 17 analyzed genomes. In contrast to findings at the interspecific level (Terakami et al., 2012), we detected identical LSC/IR and SSC/IR border regions in the 17 Ae. tauschii genomes, thereby indicating that these genomic features would be of little use as evolutionary tools for analyses at the intraspecific level. In conclusion, we found that the general structure of the 17 Ae. tauschii cp genomes including gene order and number, has been well conserved. The few variations that we detected are located in the non-coding regions (intergenic spacer regions).

Given the lower nucleotide substitution rates in cp genomes compared with nuclear genomes, it is generally assumed that a substantial sequence length is required for a robust phylogenetic analysis based on cp genome sequences (Wolfe, Li & Sharp, 1987; Khakhlova & Bock, 2006). Accordingly, in order to acquire more comprehensive information, the complete chloroplast genome sequence are beneficial for investigating the phylogenetic relationship of angiosperms (Kim et al., 2015). With respect to the origin of Ae. tauschii in China, the Yili Area of Xinjiang is undisputedly considered to represent the easternmost boundary of the natural distribution of wild Ae. tauschii population (Matsuoka, Takumi & Kawahara, 2015; Gogniashvili et al., 2016; Wang et al., 2013), whereas the origin of landraces in the Yellow River region remains a source of debate. In the study, group I was found derived from the L2 lineage, whereas the accessions in groups II and III all originated from the L1 lineage, which was consistent with the findings of analyses based on 7185 SNP markers (Wang et al., 2013), and thereby indicated that complete cp genome sequences are as equally effective as nuclear sequences for the assessment of the phylogenetic relationships among Ae. tauschii accessions. We also found that the Yellow River accessions (SC1 and T093) clustered closely in a small branch with AY22 from Pakistan (South Asia), wheras Xinjiang appeared to clustered together with Central Asian (Fig. 3). Thus, the aforementioned results imply that Ae. tauschii accessions collected from the Yellow River region (Henan and Shaanxi) could have originated from South Asian populations rather than from those in Xinjiang Province in China (Wang et al., 2010; Yen, Cui & Dong, 1984) or Iran (Wei et al., 2008), which could contribute to more effective utilization of wild germplasm resources.

Iran is believed to be the origin of Ae. tauschii in the Yellow River region according to the results of Wei et al. (2008). It is conjectured that the spreading route of qingke barley from Western Eurasia though South Asia and into Northern Tibet (Zeng et al., 2018) and the dispersal route of common wheat from the margins of the Qinghai–Tibetan Plateau to the Yangtze Valley (Wu et al., 2019) may offer valuable clues as to the introduction of Ae. tauschii in China (Betts, Jia & Dodson, 2014). In addition, some Southwest Asian domesticated accessions, including those of pea and rye, appear at Changguogou in the Yarlung Tsangpo River basin of Southern Tibet (Fu, Xu & Feng, 2000), and flax is found at Ashaonao on the southeastern Tibetan plateau (Guedes et al., 2014; Guedes et al., 2015), thereby indicating that the introduction of Ae. tauschii from South Asia is also possible. Thus, we proposed that the Yellow River accessions may have been introduced directly from South Asia.

Given the aforementioned considerations, we can tentatively propose the possible introduction route whereby Ae. tauschii dispersing eastward to China. Specifically, we believe that Ae. tauschii accessions distributed in Iran spread eastwards in two routes. One is assumed to have followed a route to Central Asia and the Yili Area of Xiinjiang through either human activities or natural extension, whereas the other one migrated eastwards, spreading to South Asia through Northern Tibet, most likely concomitant with the introduction of qingke barley into the Yellow River region, and gradually evolved into Ae. tauschii accessions found in this region today. Nevertheless, in order to verify this conjecture, it would be necessary to examine a larger number of Ae. tauschii accessions originating from sub-divided areas to facilitate a more comprehensive population analysis, and thereby enable us to trace the exact route of Ae. tauschii dispersal into China. In the present study, we investigated genetic variation of Ae. tauschii and the source of Chinese landraces, which not only provided a series of new insights to clarify the spreading route of Ae. tauschii, but also could be contribute to more effective utilization of wild germplasm resources for the genetic improvement of bread wheat.

In order to gain further insights into the origin of the D genome origin of common wheat, we we performed phylogenetic analysis based on the cp genomes of 99 accession in the Poaceae family (Fig. 4), and found that the tree we constructed is similar in topological structure to that presented by Bernhardt et al. (2017). Most researchers consider that Ae. tauschii, the donator of the D genome of common wheat, could be derived from monophyletic speciation (Dvorak et al., 1998; Dvorak et al., 2012; Wang et al., 2013; Matsuoka, Takumi & Kawahara, 2015). However, the findings of a study in which the evolutionary relationship of the A, B and D genome lineages were assessed, based on the genome sequences (2,269 genes) of hexaploid bread wheat subgenomes and five diploid relatives, indicated that the D genome is derived from homoploid hybrid speciation of the A and B genomes (Marcussen et al., 2014). On the basis of a phylogenetic analysis of chloroplast DNAs, Li et al. (2018) also conjectured that the maternal origin of the D genome lineage might be the A genome or some other relatively close lineage through ancient hybridization. In the present study, with regard to T. urartu (A genome) and Ae. speltoides (B genome), we found that Ae. tauschii firstly clustered together with the other Aegilops specie. If the D genome is derived from an ancient hybridization between the A and B genomes, Ae. tauschii should be closely clustered with T. urartu (the A genome acts a maternal parent) or Ae. speltoides (the A genome acts a maternal parent). Thus, we proposed that Ae. tauschii is derived from monophyletic speciation rather than ancient hybridization. In brief, these phylogeny results will serve as a reference framework for future studies on Triticeae or Ae. tauschii.

To identify the genetic divergence in the assembled genomes, we determined the nucleotide variability (Pi) of coding genes, intron regions and intergenic regions using DnaSP. The results revealed that the sequence divergence of the IR regions appeared to be lower than that of the LSC and SSC regions, which has also been noted in other angiosperms and may possibly be attributed to copy correction of the IR regions via gene conversion (Khakhlova & Bock, 2006). The IR sequences of 17 accessions were identical in this study. Intra-species variabilities were detected predominantly in the non-coding regions (intergenic spacer regions) of LSC and SSC (Fig. S2 and Fig. 5). We also found that four variable loci (rpl32-trnL-UAG, ccsA-ndhD, rbcL-psaI and rps18-rpl20) located in the non-coding regions showed notably high levels of nucleotide polymorphisms, two of which (rpl32-trnL-UAG, ccsA-ndhD) are located in the SSC region and the other two loci (rbcL-psaI , rps18-rpl20) are found in the LSC region (Fig. 5). Previous studies have identified Ae. tauschii accessions using the cp non-coding sequences trnF-ndhJ, trnC-rpoB, atpI-atpH, and ndhF-rpl32 (Yamane & Kawahara, 2005; Dudnikov, 2012). Here, we developed a further four marker regions (rpl32-trnL-UAG, ccsA-ndhD, rbcL-psaI and rps18-rpl20) with relatively high levels of intraspecific variation, which can be used for population genetic analyses or serve as specific DNA barcodes of Ae. tauschii (Zhang, Ma & Li, 2011; Maier et al., 1995).

Conclusions

In this study, with respect to gene order, gene number, and IR/SC boundary regions, we demonstrate that genomic structure is well conserved among the cp genomes of the 17 Ae. tauschii accessions we assembled using Illumina next-generation DNA sequencing technology. Intraspecific differences among the cp genomes were detected primarily in non-coding regions (intergenic spacer regions) which are the main features contributing to the observed differences in genome size. An analysis of the phylogenetic relationships among the accessions based on the complete genome sequences indicated that Ae. tauschii accessions from the Yellow River region in China might have originated directly from South Asia. We also confirmed that Ae. tauschii is derived from monophyletic speciation rather than hybrid speciation. Furthermore, we identified four cpDNA marker sequences (rpl32-trnL-UAG, ccsA-ndhD, rbcL-psaI, and rps18-rpl20) that can be used to study inter- and intraspecific genetic structure and diversity of Ae. tauschii.

Supplemental Information

Supplemental Information 1 The chloroplast genomes of 17 Aegilops tauschii accessions

Click here for additional data file.

Table S1 List of genes present in the Aegilops tauschii chloroplast genome

Click here for additional data file.

Table S2 The list of the Gene Ontology (GO) functional categories of the 122 functional genes

Click here for additional data file.

Table S3 The Nucleotide diversity of 17 Aegilops tauschii accessions in the sequences of coding genes, intron regions and intergenic regions

Click here for additional data file.

Table S4 The used primers of four variable genomic regions

Click here for additional data file.

Figure S1 The Gene Ontology (GO) functional categories of the 122 functional genes

Click here for additional data file.

Figure S2 Sequences alignment of the chloroplast genomes between T093 (L1 lineage) and AY81 (L2 lineage) using AL8/78 as the reference genome

The vertical scale of mVISTA visualization image denotes the percentage of identity, ranging from 50% to 100%. The horizontal axis marked color code as protein-coding genes, rRNA, tRNA or conserved non-coding regions.

Click here for additional data file.

Fgirue S3 Positions of IR-LSC and IR-SSC boundary in 17 Aegilops tauschii chloroplast genomes

Click here for additional data file.

Additional Information and Declarations

Competing Interests

Author Contributions

Data Availability

The authors declare there are no competing interests.

Qing Su conceived and designed the experiments, performed the experiments, analyzed the data, prepared figures and/or tables, authored or reviewed drafts of the paper, and approved the final draft.

Luxian Liu conceived and designed the experiments, analyzed the data, prepared figures and/or tables, authored or reviewed drafts of the paper, and approved the final draft.

Mengyu Zhao performed the experiments, analyzed the data, authored or reviewed drafts of the paper, and approved the final draft.

Cancan Zhang performed the experiments, prepared figures and/or tables, and approved the final draft.

Dale Zhang performed the experiments, analyzed the data, prepared figures and/or tables, authored or reviewed drafts of the paper, and approved the final draft.

Youyong Li and Suoping Li conceived and designed the experiments, authored or reviewed drafts of the paper, and approved the final draft.

The following information was supplied regarding data availability:

Chloroplast genome sequences are available at GenBank: MN258085 (SC1); MN258083 (AY81); MN258078 (AY34); MN223978 (AY22); MN258084 (AY320); MN223977 (AY21); MN258087 (XJ04); MN258089 (XJ0109); MN258086 (T093); MN258088 (XJ098); MN258082 (AY78); MN223975 (AS060); MN258080 (AY48); MN258090 (AY076); MN223976 (AY20); MN258081 (AY72); MN258079 (AY46).

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
