# Peer review of "The complete chloroplast genomes of seventeen Aegilops tauschii: genome comparative analysis and phylogenetic inference"

_PeerJ, doi:10.7717/peerj.8678_

## Round 0.1 · original submission · Major Revisions

Thank you for your contribution. The manuscript has been reviewed by two reviewers and there appear to be several sections where clarity needs improvement; this impedes getting the correct message across. It would help to have assistance with use of a proofreading and editing service. However, some of the preliminary reviews appear to have valuable content which should be considered for your revisions.

As an additional note, journal manuscripts are often scanned by text-mining software that locates and extracts core data elements, like gene function. Adding standard ontology terms, such as the Gene Ontology (GO, geneontology.org) or others from the OBO foundry (obofoundry.org) can enhance the recognition of your contribution and description. This will also make human curation of literature easier and more accurate. GO: annotations help to define the biological, molecular, and cellular context of the gene.

This manuscript will be placed in a ‘major revision’ category until revisions can be made. Thank you for your contribution. We hope to receive your revisions soon.

Reviewer 1 ·

Basic reporting

The paper presents relevant data for the field of wheat improvement. The writing needs considerable improvement beyond the duty of a peer-review editor. I attach a copy of my review with notes on the first few pages only.

Some figures need to be improved:
Figure 1, some of the text in the figure is unreadable
Figure 4, the subcategories for each of the gene are not distinguishable.
Figure 5, the labels are unreadable because of low resolution/quality.

I tried to retrieve the accessions from GenBank but they are not available.

Experimental design

The experimental design is appropriate. The authors surveyed a relatively large number of accessions representing the native range of the species.

Validity of the findings

The conclusions reached by the authors are supported by the data

Annotated reviews are not available for download in order to protect the identity of reviewers who chose to remain anonymous.

Reviewer 2 ·

Basic reporting

The paper needs to be text edited in multiple sections.

For instance, in the abstract the authors wrote " IR expansion and contraction with identical structure among 17 Aegilops tauschii accessions were not influence chloroplast genomes in length. " (not sure what the authors mean in this sentence).
And also "The ndhF gene in AY46 accession appeared the highest ω value". (probably showed or had instead of appeared").
And also "The specific spreading route of Ae. tauschii revealed in this work, reflects the frequent cultural exchange through the silk road from one point of view." (not sure what the authors mean by "one point of view").

Examples like this occur across the manuscript and really constrain the all message of the paper. Figures and tables appear to be good but note that population labels of Figure 1 are very hard to see. Please consider increasing labels and changing colours.

Raw data is shared. Thank you for that!

Experimental design

The aims and experimental design appear to be rigorous but as stated before, the language is sometimes ambiguous. Methods are clear.

Validity of the findings

The Discussion needs to more clear. I don't really understand the main conclusions of this paper. For instance:

- Because the ndhF gene had the highest ω value, the authors conclude that with might be involved in the adaptation to high altitude ecological environment. This is quite speculative. It might also be involved in something else. Note that this gene evolves at a very fast rate in most plants.

- The phylogenetic relationships support that Ae. tauschii in the Yellow River region might be directly originated from Central Asia rather than Xinjiang. Please explain this better.

- The authors also conclude that "The specific spreading route of Ae. tauschii revealed in this work, reflects the frequent cultural exchange through the silk road from one point of view. We confirmed that Ae. tauschii derived from monophyletic speciation rather than hybrid speciation at the chloroplast genome level." I'm really lost in these two sections. First, I don't see how the authors can link this to the silk road. Perhaps, the authors are right but they need to explain this much better. But from the current text/figures/tables, I cannot really support this conclusion. The second sentence, is simply wrong. There is no result in this study that can confirm or reject the presence of hybrid speciation, and even less at the chloroplast level (unless the authors can sustain that the chloroplast is biparentally inherited in Ae, and in that case they have to re-write previous conclusions).

Additional comments

This study has some potential. It appears to be methodologically solid but there are too many unclear and ambiguous sentences that prevents me from accepting this manuscript. I'm also worried about the conclusions drawn by the authors. I hope my suggestions help them to prepare a new version.

---

## Round 0.2 · Minor Revisions

Many reviewers of chloroplast genome papers wish to see significant differences highlighted rather than see how similar an extended study results in. Unfortunately, you indicate that the IR regions are identical, but you do make some mention of inter-species variability. This point is merely mentioned and not fully explained with concrete data comparisons or offerings. If indeed there are difference, providing the raw data in a rar format does not lead the reader to discovering the differences, but makes the reader need to re-evaluate the data o there own. In some context the reader should be made aware of the differences; non of the other supplementary files appear to offer this. It is disappointing to see a section highlight sequence analysis and marker development; but, only figures are presented with no usable sequence data form to work with. I would imagine that a goal in developing the molecular markers would be so that a scientist would be able to use probes to assess the etiology of their own germplasm; no data is presented for this to happen.

To perhaps add additional value, a definition for the gene set, especially the 127 functional genes might be more valuable if annotations would be assigned.

Journal manuscripts are often scanned by text-mining software that locates and extracts core data elements, like gene function. Adding standard ontology terms, such as the Gene Ontology (GO, geneontology.org) or others from the OBO foundry (obofoundry.org) can enhance the recognition of your contribution and description. This will also make human curation of literature easier and more accurate.
None of this was visible.

I would suggest that additional enhancements are still required, and upon return perhaps another review may be required. I apologize that the findings were not better; however, there has been recent requirements to heighten the evidence required to bring new data to light when plastid organelle data is involved. I understand that the Aegilops species are key elements of the bread wheat origins, but the context and application of the study is not clear at this point. I will place the manuscript at minor revision until additional justifications can be made.

---

## Round 0.3 · Minor Revisions

The annotations in Figure S1 lacks clarity (terms truncated). The link of sequences to the actual GO: number annotations will link the verbose term to the searchable GO: term. Please add these in some fashion to the manuscript. The manuscript should be in fine form to accept once this is done. Thank you for your attention.

---

## Round 0.4 · accepted · Accept

There was mention of adding the GO terms contained within the manuscript; however, from my access point I was not able to view the supplemental files. The read was fairly clean with a few exceptions mentioned below. The purpose of the chloroplast study is defined well, and from the sampled accessions there was data provided to propose a hypothesis for the spread of species genotypes. Considering that there has been back and forth discussion with this manuscript I will agree that it is now in an acceptable form for publication. The approach at annotating the diversity may have utility in categorizing germplasm for future uses. Thank you for addressing the concerned presented to you; I will suggest that this manuscript be moved forward. Thank you for your efforts. There were a few needed edits seen:
EDITS
LINE NO: / BEFORE / AFTER / [COMMENTS]
LINE 47: / could be contribute to more / . / [?]
LINE 68: / showes / . / [.]